# Evaluation of the Effect of a Combined Chemical and Thermal Modification of Wood though the Use of Bicine and Tricine

Dennis Jones [1,2,*], Davor Kržišnik [3], Miha Hočevar [3], Andreja Zagar [3], Miha Humar [3], Carmen-Mihaela Popescu [4,5], Maria-Cristina Popescu [4], Christian Brischke [6], Lina Nunes [7], Simon F. Curling [8], Graham Ormondroyd [8] and Dick Sandberg [1,2]

[1] Wood Science and Engineering, Luleå University of Technology, Forskargatan 1, SE-931 87 Skellefteå, Sweden; dick.sandberg@ltu.se
[2] Department of Forestry and Biomaterials, Czech University of Life Sciences Prague, 16500 Prague, Czech Republic
[3] Biotechnical Faculty, University of Ljubljana, Jamnikarjeva 101, 1000 Ljubljana, Slovenia; davor.krzisnik@bf.uni-lj.si (D.K.); miha.hoc@gmail.com (M.H.); andreja.zagar@bf.uni-lj.si (A.Z.); miha.humar@bf.uni-lj.si (M.H.)
[4] Petru Poni Institute of Macromolecular Chemistry of the Romanian Academy, 41A Grigore Ghica Voda Alley, 700487 Iasi, Romania; mihapop@icmpp.ro (C.-M.P.); cpopescu@icmpp.ro (M.-C.P.)
[5] Centre of Wood Science and Technology, Edinburgh Napier University, Edinburgh EH11 4EP, UK
[6] Wood Biology and Wood Products, University of Goettingen, Buesgenweg 4, D-37077 Goettingen, Germany; christian.brischke@uni-goettingen.de
[7] Structures Department, LNEC, National Laboratory for Civil Engineering, Av. do Brasil, 101, 1700-066 Lisbon, Portugal; linanunes@lnec.pt
[8] The Biocomposites Centre, Bangor University, Deiniol Road, Bangor LL57 2UW, UK; s.curling@bangor.ac.uk (S.F.C.); g.ormondroyd@bangor.ac.uk (G.O.)
* Correspondence: dennis.jones@ltu.se

**Abstract:** The effects of thermal modification of wood have been well established, particularly in terms of reductions in mechanical performance. In recent years, there has been an increase in studies related to the Maillard reaction. More commonly associated with food chemistry, it involves the reaction of amines and reducing sugars during cooking procedures. This study has attempted to combine the use of amines and thermal modification, with subsequent properties investigated for the treatment of spruce (*Picea abies* (L.) H. Karst) and beech (*Fagus sylvatica* L.). In this initial study, the combined effects of chemical treatments by tricine and bicine were investigated with thermal modification. Along with some preliminary data on mechanical properties, the modifications which appeared in the wood structure were evaluated by infrared spectroscopy and biological studies according to EN113 and EN117 methodologies. The hierarchal study interpretation of FTIR suggested interactions between the bicine or tricine and the wood, which was partly supported by the analysis of volatile organic compounds (VOC), though other tests were not as conclusive. The potential of the method warrants further consideration, which will be described.

**Keywords:** wood; Maillard reaction; thermal/chemical treatment; mechanical properties; infrared spectroscopy; biological properties

## 1. Introduction

Wood has long been associated as an important natural resource, given its use by humanity to produce such diverse items as art and furniture and its use in construction. The use of wood is seen as a necessity for sustainable construction in our modern society [1]. As greater importance is placed upon the performance of wood as a material, it has become necessary to alter some of the inherently undesirable properties limiting the longevity of service performance. Of particular importance are issues relating to limitations related to stability in service, susceptibility to fungal decay or weathering, etc., [2,3]. Thus, ways of

improving factors such as moisture sensitivity, low dimensional stability, low hardness and wear resistance, low resistance to bio-deterioration against fungi, termites, marine borers and low resistance to UV radiation have become important parameters in wood treatment technologies and in particular wood modification treatments.

Nowadays, wood modification is defined as a process adopted to improve the physical, mechanical or aesthetic properties of sawn timber, veneer or wood particles used in the production of wood composites. This process produces a material that can be disposed of at the end of a product's life cycle without presenting any environmental hazards greater than those that are associated with the disposal or combustion of unmodified wood. Recent publications [4,5] indicated how wood modification is becoming more established across Europe at both the commercial large and the niche manufacturing scale. A range of methods have become available, with the focus on thermal [6–9] and chemical treatments [10–13] being the most studied. The subject has attracted several reviews, e.g., [14–16], and is hereby introduced in more detail.

The thermal modification of wood is now an accepted means of treating wood, with several commercialised processes. Depending on the process, humidity, temperature and time [17], a range of reactions may occur, such as hydrolysis, oxidation and decarboxylation reactions, along with physical changes. The results of such treatments are end products presenting improved physical characteristics, such as lower hygroscopicity, better dimensional stability and durability, though mechanical properties are often significantly reduced, with dynamic mechanical properties (such as impact bending, fatigue) typically being more affected than static properties. Improvements in stability and durability are of importance where wood is exposed to different chemicals or biological agents such as fungi and bacteria, or to frequent use under natural environmental conditions.

Another possible type of wood modification is the reaction with different chemicals—chemical modification—where chemical moieties are covalently bonded to the wood cell wall polymers. Different chemical modifications [11–13,18–20] may be used to alter the properties of wood [20], particularly for improving the dimensional stability, decay resistance and water performance of wood, and to improve mechanical properties compared to those from typical thermal modification processes [21]. Most modification methods involve the hydroxyl groups in the cell wall, which are partially substituted, and the cell wall of the wood is bulked with the bonded chemicals. The substitution of the hydroxyl groups reduces the number of primary sorption sites (typically the OH groups), while the bulking reduces the volume in the wood cell wall which is available to water molecules. The most common chemical modification process involved reaction with anhydrides, with a multitude of examples listed in the literature. However, it is the acetylation process using acetic anhydride that has been the most studied, resulting in its commercialisation [2]. Apart from anhydrides, many other chemicals have been used to try to improve the properties of wooden material. For example, treatments with dimethyloldihydroxyethylene urea (DMDHEU), melamine resin, silane or silicon polymers were found to improve the mechanical strength of wood [3].

More recently, the use of Maillard reactions for wood treatment proved to be a promising modification method to improve some of the wood properties. This type of reaction is well-known in food chemistry, where it is responsible for the browning in many foods during baking [22,23]. The essence of the reaction is that a reducing sugar condenses with a compound possessing a free amino group to give a condensation product [24]. Subsequently, a range of reactions takes place, including cyclisations, dehydrations, retroaldolisations, rearrangements, isomerisations and further condensations, which ultimately lead to the formation of polymers and co-polymers, known as melanoidins [24]. The composition of its chemical structure is relatively unknown due to the complexity of the products that are generated in the reaction [25]. The advantage of this reaction is that it is an aqueous process and initiated by heat only, making it relatively straightforward to apply to wood in a commercial process. In addition, the reaction does not require the use of strong acids or bases, which could degrade the wood structure.

In earlier experiments [26], the influence of the Maillard reaction on European beech and Scots pine was investigated. The wood was impregnated with an amine (glucosamine, lysine or glycine), sugars (glucose or xylose) and an extra reagent to improve the reaction (magnesium chloride, maleic acid or citric acid). The results showed that when lysine, glucose or citric acid were reacted, a high weight percentage gain (WPG) was obtained (18% for beech and 40% for pine). After a leaching procedure, the WPG for beech was 11% and 25% for pine, respectively. This preliminary screening reaction has shown that the Maillard reaction offers potential as a potential new wood modification system, though a reduction in leaching would help maintain the modification benefits.

In this context, a recent work of Hauptmann and co-workers [3] considered the use of tricine for the modification of wood. This is capable of binding with reduced sugars, though studies have been limited to a maximum temperature of 103 °C. They observed increased hardness and the tensile strength of their modified wood species. As indicated earlier, Maillard type reactions are a cascade of reactions (known also as nonenzymatic browning). The initial step consists in a reaction between a reducing end of a saccharide and an amino acid [3,26]. This reaction takes place usually during the heat processing of food with a low relative humidity [27] and it is considered to cause the brown stains during wood kiln drying with temperatures higher than 80 °C due to the presence of natural amino acids and reduced sugars [3]. The most efficient characterisation method to identify the modifications appearing in the wood structure during heating and to identify the possible interaction and bonds formed between the wood components and the reagents used in chemical modification is infrared spectroscopy. This technique proved to be an efficient tool to identify small modifications in the wood structure appearing during different treatments [17,28,29] such as decay, i.e., [30,31] or photodegradation, i.e., [32,33]. Moreover, due to the use of the small amount of sample and little or no processing, this method is considered by many to be non-destructive.

In this study, the combined effects of chemical treatment and the thermal modification are investigated in terms of the use of bicine [2-(Bis (2-hydroxyethyl) amino) acetic acid] and tricine [N-(2-Hydroxy-1,1-bis (hydroxymethyl)ethyl) glycine] (Figure 1). The concept of the Maillard reaction depends upon the availability of an active site on the nitrogen within the amino group (i.e., a N-H group). Whilst this is not present in bicine, it was postulated within this work that the compound may undergo side group displacement, allowing the general concepts within the Maillard reaction to proceed. Some of the resulting properties for the treatment of beech are considered (mechanical, physical and bioresistivity), along with infrared spectroscopic analysis in an attempt to demonstrate actual chemical bonding.

**Figure 1.** Chemical structure of bicine (**a**) and tricine (**b**).

## 2. Materials and Methods

In this study, the potential of combining chemical and thermal treatments for European beech (*Fagus sylvatica* L.) and Norway spruce (*Picea abies* (L.) H. Karst) was determined. For both species, material sourced and felled in Slovenia was used. For each treatment set, separate sets without visible growth features and varying ring widths, mainly in semi-radial orientation, were prepared from heartwood sections. Thus, only mature wood was used for these experiments. The density ranges of the samples were within the typical oven-dry ranges commonly noted in the scientific literature. Thus, bicine (Bi, CAS Number 150-25-4, Fisher Scientific, Waltham, MA, USA) and tricine (Tri, CAS Number 5704-04-1,

Fisher Scientific, Waltham, MA, USA) were impregnated into wood specimens, using a 10% solution (weight:volume, *w:v*) of the respective compound per impregnation.

### 2.1. Specimen Treatment

Beech (hereafter referred to as B) and spruce (hereafter referred to as S) specimens of varying dimensions depending on subsequent testing were prepared and subjected to the following treatment regimes (Table 1).

**Table 1.** Overview of treatments undertaken.

| Species | Code | Treatment Description |
|---------|------|----------------------|
| Beech | B_HT | Heat treatment only |
| | B_Bi | Bicine pre-treatment and drying |
| | B_Bi_HT | Bicine pre-treatment and heat treatment |
| | B_Tri | Tricine pre-treatment and drying |
| | B_Tri_HT | Tricine pre-treatment and heat treatment |
| | B_C | Control |
| Spruce | S_HT | Heat treatment only |
| | S_Bi | Bicine pre-treatment and drying |
| | S_Bi_HT | Bicine pre-treatment and heat treatment |
| | S_Tri | Tricine pre-treatment and drying |
| | S_Tri_HT | Tricine pre-treatment and heat treatment |
| | S-C | Control |

For the impregnation procedure, the solutions were introduced using a vacuum-pressure impregnation (VPI) process according to the full-cell process in a laboratory impregnation setup (Kambič, Semič, Slovenia). It consisted of 30 min of vacuum ($1.0 \times 10^4$ Pa), 40 min of pressure ($10 \times 10^5$ Pa) and 10 min of vacuum ($1.5 \times 10^4$ Pa). Reagent uptakes were subsequently determined gravimetrically. The impregnated specimens were conditioned (23 °C; 65% relative humidity (RH)) for 2 weeks prior to thermal modification. Non-impregnated and conditioned specimens served as controls.

The thermal modification (HT) was performed according to a modified Silvapro® commercial procedure [34] limited to 165 °C, due to the risk of thermal degradation of the chemicals if exposed to temperatures above 180 °C. Control specimens were only heated up to 100 °C during the drying procedure in atmospheric conditions. The time of thermal modification at the target temperature was 3 h and mass loss (ML) of the specimens after thermal modification was determined gravimetrically. The HT specimens were stored in the laboratory for 4 weeks (23 °C; 65% RH) before subsequent testing.

### 2.2. Physical Tests

#### 2.2.1. Colour Analysis

Colour was determined on the semi-radial surfaces of a selection of specimens. The colour measurements were performed according to the CIE L*a*b* system, a method created by the Commission International de l'Eclairage. The CIE L*a*b* system is characterised by three parameters: L*, a* and b*. The L* axis represents the lightness, which varies from a hundred (white) to zero (black), representing the achromatic axis of greys, whereas a* and b* are the chromaticity coordinates. A positive value of a* denotes a redder colour on a green–red scale, whereas a positive value of b* denotes a more yellow colour on a blue–yellow scale. Together, those three components form a three-dimensional colour space. Colour measurements of in-service testing were performed several times a year with a portable Colour Measuring Device (EasyCo 566, Erichsen, Hemer, Germany) and expressed in the CIE L*a*b* system. This device enables contact-free precise colour measurement. The diameter of the measurement spot is 20 mm. However, laboratory test specimens were scanned and processed with Corel Photo-Paint 8 software. Corel Photo-Paint was used as colour analysis as this technique provides the colour of the whole surface and not of

individual spots. Total colour difference ΔE* (Equation (1)) from a reference colour (L*0, a*0, b*0) to a target colour (L*1, a*1, b*1) in the CIE Lab space is calculated by determining the Euclidean distance between two colours given by:

$$\Delta E = ((\Delta L^*)^2 = (\Delta a^*)^2 + (\Delta b^*)^2)^{0.5} \tag{1}$$

2.2.2. Contact Angle Measurements

Contact angles were detected using a Theta optic tensiometer (Biolin Scientific Oy, Espoo, Finland) and OneAttesion 2.4 (r4931) software (Biolin Scientific, Espoo, Finland). Five replicates of each treatment were used, to which were added 2 droplets of water (4 μL each). The samples used were 25 × 15 × 50 mm in size, having been thoroughly dried (103 °C, 24 h). Subsequent changes in contact angle were determined using a 7.6-megapixel camera, with software constantly monitoring subsequent wetting over a period of 60 s.

2.2.3. Dynamic Vapour Sorption (DVS) Analysis

Samples for dynamic vapour sorption (DVS) analysis were milled in a Retsch SM 2000 cutting mill (Retsch GmbH, Haan, Germany) with a Conidur® perforation sieve with 1.0 mm perforations. Thus, several samples were milled together to create an average mix of fibres representing each treatment. The milled wood samples were conditioned at 20 ± 0.2 °C and 1 ± 1% RH through blowing with dry air. Analysis of the wood samples was performed using a DVS apparatus (DVS Intrinsic, Surface Measurement Systems Ltd., London, UK). A small amount (approximately 400 mg) of pre-conditioned wood chips was placed on the sample holder, which was suspended in a microbalance within a sealed thermostatically controlled chamber, in which a constant flow of dry compressed air was passed over the sample at a flow rate of 200 cm$^3$/s and a temperature of 25 ± 0.2 °C. The schedule for DVS had two steps: 0% and 95% RH. The DVS maintained a given RH until the weight change of the sample was less than 0.002%/min for at least 10 min. The running times, target RH, actual RH and sample weights were recorded three times per min throughout the isotherm run. After the DVS analysis, equilibrium moisture content at 95% RH (EMC95% RH) was determined along with the hysteresis values, determined as the difference between desorption and adsorption values at a given RH.

*2.3. Mechanical Tests*

2.3.1. Mechanical Performance Tests

Modulus of elasticity (MOE) and modulus of rupture (MOR) were determined according to EN 310 [35] with a static three-point bending test on a Zwick Z005 universal testing machine (Zwick-Roell, Ulm, Germany). In total, 60 specimens (10 replicates for each tested material) with dimensions 360 × 20 × 20 mm were prepared and oven dried (103 °C) until a constant mass was achieved. They were dried in order to eliminate the influence of different moisture contents between control and modified wood. The specimens were tested for MOE and MOR immediately after drying.

Compressive strength was determined according to the ASTM D1037-99 standard [36] on a Zwick Z100 universal testing machine (Zwick-Roell, Ulm, Germany). In total, ten specimens for each test group with dimension 50 × 20 × 20 mm were prepared and oven dried until a constant mass was achieved. The specimens were tested for compressive strength immediately after conditioning under standard conditions. After the test, the compressive strength ($F_m$) was calculated.

The Brinell hardness (HB) was determined by a standard test method according to EN 1534 [37]. The penetration depth (h) of an iron sphere (D = 10 mm) was used for 5 replicates, each undergoing 4 individual measurements, in calculations at load F = 1000 N to determine the Brinell hardness according to:

$$HB = \frac{2 \cdot F}{\pi \cdot D \cdot \left( D - \sqrt{\left( D^2 - 4 \cdot h \cdot (D - h) \right)} \right)} \tag{2}$$

### 2.3.2. High Energy Multiple Impact (HEMI) Test

The development and optimisation of the high energy multiple impact (HEMI) test has been previously described by Rapp et al. [38]. For testing modified and non-modified control specimens in a high-energy multiple impact (HEMI) test, specimens of $20 \times 20 \times 10$ mm were split in four specimens of $5 \times 20 \times 10$ mm (radial $\times$ tangential $\times$ longitudinal). Five times, n = 20 samples of $5 \times 20 \times 10$ mm were submitted to the HEMI tests. HEMI tests were performed in a heavy-impact ball mill (Herzog HSM 100-H; Herzog Maschinenfabrik, Osnabrück, Germany). In short, 20 oven-dried (103 °C) samples were placed in the bowl (140 mm in diameter) of the mill, together with one steel ball of 35 mm in diameter plus three balls of 12 mm and 6 mm in diameter, respectively. For crushing the specimens, the bowl was shaken for 60 s at a rotary frequency of 23.3 $s^{-1}$ and a stroke of 12 mm. The fragments of the 20 specimens were fractionated on a slit sieve according to EN ISO 5223 [39] with a slit width of 1 mm using an orbital shaker at an amplitude of 25 mm and a rotary frequency of 200 $min^{-1}$ for 2 min. The degree of integrity (I), fine percentage (F) and resistance to impact milling (RIM) were calculated following Equations (3)–(5), where $m_{20}$ is the oven-dry mass of the 20 biggest fragments, $m_{all}$ is the oven-dry mass of all fragments and $m_{fragments<1mm}$ is the oven-dry mass of fragments smaller than 1 mm. The RIM was calculated according to the optimised method [38], which contains a three-fold weighting of the fine fraction ($3 \times F$), with the values 300 and 400 guaranteeing to achieve a maximum RIM value of 100%.

$$I = m_{20}/m_{all} \tag{3}$$

$$F = m_{fragments<1mm}/m_{all} \tag{4}$$

$$RIM = ((I - 3 \times F) + 300)/400 \tag{5}$$

### *2.4. Chemical Tests*

### 2.4.1. Volatile Organic Compound (VOC) Analysis

In order to aid FTIR analysis and attempt to identify Maillard-type reaction products, it was decided to undertake gas chromatography–mass spectrometry (GCMS) analysis of samples. Thus, subsamples of the materials were cut and immediately placed into a minichamber (Markes International, Llantrissant CF72 8XL, UK). A flow of nitrogen at 2 mL/min was passed over the samples and through Tenax columns to collect any volatile organic compounds (VOCs) released. Samples were kept in the chamber for 2, 8 or 16 h. Collected VOCs were eluted form the Tenax columns using methanol and analysed using an electron impact (EI) capillary GCMS (Glarus 680 gas chromatograph, Perkin Elmer, using an Agilent VF5-MS column (30 m $\times$ 0.25 mm $\times$ 0.25 μm), coupled with a Clarus 600 C mass spectrometer, Perkin Elmer). GC conditions were an initial temperature of 60 °C for 1 min, followed by a temperature increase of 6.0 °C/min to a maximum temperature of 300 °C, with the final temperature being held for a further 10 min. Mass spectra were collected between elution times of 3–51 min from the gas chromatograph over an electron impact mass range of 40.00 to 600.00.

### 2.4.2. Infrared Spectroscopy

The infrared spectra of the reference and treated wood specimens were recorded in potassium bromide (KBr) pellets on a Bruker ALPHA FT-IR spectrometer with 4 $cm^{-1}$ resolution. The concentration of the sample was a constant of 2 mg/200 mg KBr. Processing of the spectra was performed using the Grams 9.1 program (Thermo Fisher Scientific, Waltham, MA, USA).

### *2.5. Effects against Biological Deterioration Tests*

In order to ascertain any biological effects of the treatments, the following aspects were evaluated within this section.

2.5.1. Resistance to Fungal Decay

In order to determine any potential effectiveness of treatment against wood-destroying fungi, tests were carried out according to EN113 [40] once samples had undergone artificial ageing to remove possible extractives and other components that could act like surfactants. A similar process occurs in nature with the first extensive rain period. The European EN 84 standard [41] describes a method for artificial ageing (leaching) of wood before testing the biological effectiveness. This standard was designed to simulate extensive leaching by natural precipitation. The first step was impregnation with demineralised water. The samples were stacked in a container, weighed down and vacuum impregnated (4 kPa) with demineralised water for 20 min and soaked for an additional 2 h. Samples were then immersed in water for 14 days and during this period water was replaced nine times. After the ageing process was completed, samples were dried in ambient conditions for 2 weeks prior to subsequent tests.

The decay test was performed according to the modified EN 113 [40] on treated and untreated beech and spruce samples. Disposable Petri dishes containing 20 mL of 4% potato dextrose agar (PDA, Difco, NJ, USA) were inoculated with 3 different fungi: one white rot fungi (*Trametes versicolor* (L.) Lloyd (ZIM L057)) and two brown rot fungi: *Gloeophyllum trabeum* (Pers.) Murrill (ZIM L018) and *Poria monticola* Murrill (ZIM L033). The fungal isolates originate from the fungal collection of the Biotechnical Faculty, University of Ljubljana, Slovenia. A plastic mesh was used to avoid direct contact between the samples and the medium. The assembled test dishes were then incubated at 25 °C and 80% RH for 12 weeks.

Samples of dimensions 8 mm × 25 mm × 25 mm were prepared and 5 replicates per fungus species were used for each type of treatment. The untreated beech and spruce wood samples served as reference wood species to assess the validity of the test. After incubation, the fungal mycelium was removed and the samples weighed for moisture content. After 24 h of drying at 103 °C, mass loss was determined gravimetrically.

2.5.2. Efficacy against Subterranean Termites

Subterranean termites belonging to the species *Reticulitermes grassei* Clément were captured in a pine forest in Sesimbra, Setúbal district of Portugal and were brought to the laboratory and kept in a conditioned room at 24 ± 1 °C and 8 ± 5% RH. Groups of 150 workers of termites were established in 200 mL glass jars with moistened sand (Fontainebleau sand and water; 4:1 *v*/*v*) as substrate. Three replicates (30 × 10 × 10 mm) per treatment were then placed in contact with the termites and the test run for four weeks at the described conditions. Maritime pine test specimens with the same dimensions were also included as internal virulence controls [42,43]. The initial moisture content of the blocks was measured in sets of three additional replicates per treatment and these values were used to determine the theoretical initial dry mass (IDM) of the exposed specimens (in all tests conducted). At the end of the trial, the final moisture content was recorded and the mass loss was obtained according to:

$$\text{Mass loss} = (\text{FDM} - \text{IDM})/\text{IDM} \qquad (6)$$

where FDM is the oven-dry mass of the block at the end of the test. The survival of the termites was also recorded and all wood blocks were graded in terms of termite attack using the scale: 0 = no damage; 1 = attempted attack; 2 = slight damage; 3 = superficial and inner damage; 4 = heavy inner damage.

*2.6. Statistical Analysis*

Where applicable, a multivariate analysis of variance (ANOVA) was performed to determine any significant affect as a result of specific treatments and resulting tests. Significance established at the $p < 0.05$. Tukey's honest significance test was applied to find

means that are significantly different from each other. Additionally, statistical analysis was undertaken using IBM SPSS Statistics V26.

## 3. Results and Discussion

### 3.1. Weight Uptakes

Figure 2 gives an overview of the weight changes as a result of the treatments undertaken. The results from experiments involving spruce produced weight gains from treatment with bicine or tricine, respectively. However, the combined chemical treatment of bicine with thermal treatment resulted in what appears to be an erroneous higher weight gain than that observed for treatment with bicine alone, though standard deviation could suggest similar results for S_Bi and S_Bi_HT, respectively. Since the thermal modification process resulted in a weight loss, it would be expected that similar thermal treatment following chemical treatment would result in some reduction in weight uptake unless some degree of interaction of bicine with components that would normally be lost during the thermal modification has occurred. The mass loss noted after thermal modification (1.30%) can be explained by the mild conditions applied during the experiments. This would seem to suggest that there is some retention of components because of the thermal process, which can react with the bicine. This contradicts the expected Maillard-type reaction, which is dependent on the presence of a reactive N-H group, usually in a primary amine, but suggested in work by Hauptmann et al., [3] as being possible in the secondary amine of tricine.

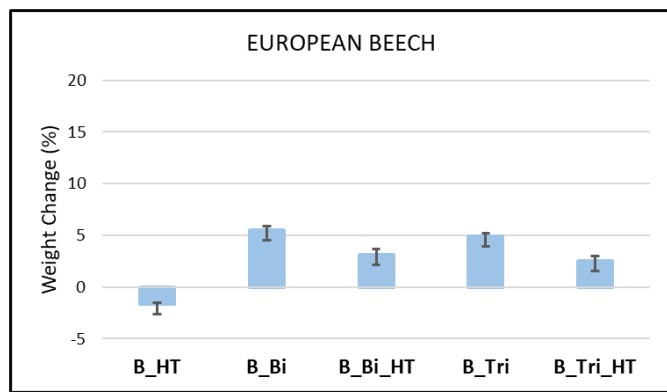 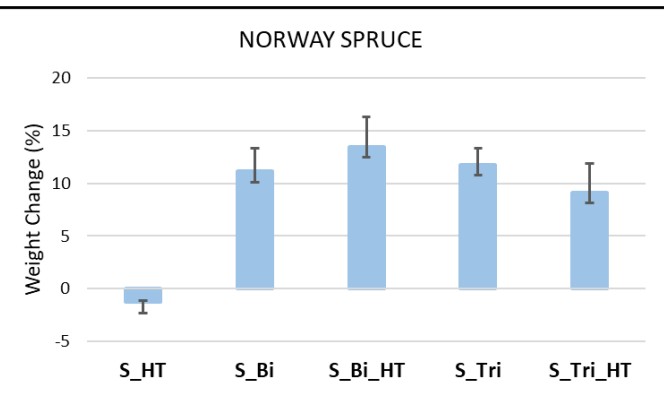

**Figure 2.** Weight changes as a result of treatments on beech (B) and spruce (S). Bi = bicine, Tri = tricine, HT = thermal modification.

### 3.2. Results from Physical Tests

#### 3.2.1. Colour Changes

Evaluation of the colour changes as a result of the treatments can be seen in Tables 2 and 3 for European beech and Norway spruce, respectively.

The results for spruce indicated that treatment with bicine in conjunction with heat treatment had the greatest impact in terms of darkening ($\Delta L^*$) of samples, resulting in a value of $-30.4$. However, with beech, it was the corresponding treatment with tricine that produced the greatest darkening ($\Delta L^* = -23.04$), compared to the darkening by heat treatment alone, where $\Delta L^*$ values of $-12.4$ and $-11.1$ were obtained for bicine and tricine, respectively. For spruce, $\Delta L^*$ was lower for tricine with heat treatment ($-15.7$), but still significantly different from the heat treatment alone. Similar effects were noted for bicine when heat treated with beech. In general, a* and b* values remained relatively unchanged for all samples.

**Table 2.** Colour determinations of European beech (B) treated with bicine (Bi) and tricine (Tri), respectively. HT = thermal modification.

|  | B_C | B_HT | B_Bi | B_Bi_HT | B_Tri | B_Tri_HT |
|---|---|---|---|---|---|---|
| L* (s.d.) | 76.00 (2.59) | 64.94 (0.96) | 64.86 (0.96) | 63.14 (1.36) | 64.72 (2.22) | 52.96 (3.26) |
| a* (s.d.) | 7.68 (0.18) | 6.90 (0.27) | 9.70 (0.27) | 8.02 (0.32) | 7.86 (0.62) | 9.3 (0.60) |
| b* (s.d.) | 8.90 (0.22) | 7.68 (0.19) | 10.20 (0.27) | 8.20 (0.27) | 7.94 (0.37) | 8.40 (0.22) |
| ΔL* |  | −11.06 | −11.14 | −12.86 | −11.28 | −23.04 |
| Δa* |  | −0.78 | 2.02 | 0.34 | 0.18 | 1.66 |
| Δb* |  | −1.22 | 1.30 | −0.70 | −0.96 | −0.50 |
| ΔL* (%) |  | −14.55 | −14.66 | −16.92 | −14.84 | −30.32 |
| Δa* (%) |  | −10.16 | 26.30 | 4.43 | 2.34 | 21.61 |
| Δb* (%) |  | −13.71 | 14.61 | −7.87 | −10.79 | −5.62 |
| ΔE* |  | 11.15 | 11.40 | 12.88 | 11.32 | 23.11 |

**Table 3.** Colour determinations of Norway spruce (S) treated with bicine (Bi) and tricine (Tri), respectively. HT = thermal modification.

|  | S_C | S_HT | S_Bi | S_Bi_HT | S_Tri | S_Tri_HT |
|---|---|---|---|---|---|---|
| L* (s.d.) | 84.46 (1.08) | 72.06 (1.33) | 75.20 (1.96) | 54.04 (1.61) | 83.26 (0.22) | 68.80 (1.61) |
| a* (s.d.) | 6.90 (0.45) | 8.20 (0.27) | 8.40 (0.55) | 9.52 (0.38) | 7.00 (0.22) | 8.20 (0.41) |
| b* (s.d.) | 10.78 (0.38) | 11.80 (0.35) | 13.60 (0.42) | 10.98 (0.18) | 13.06 (0.40) | 13.02 (0.40) |
| ΔL* |  | −12.40 | −9.26 | −30.42 | −1.20 | −15.66 |
| Δa* |  | 1.30 | 1.50 | 2.62 | 0.10 | 1.30 |
| Δb* |  | 1.02 | 2.82 | 0.20 | 2.28 | 2.24 |
| ΔL* (%) |  | −14.7 | −11.0 | −36.0 | −1.40 | −18.5 |
| Δa* (%) |  | 18.8 | 21.7 | 38.0 | 1.40 | 19.1 |
| Δb* (%) |  | 9.50 | 26.20 | 1.90 | 21.20 | 6.90 |
| ΔE* |  | 12.51 | 9.80 | 30.53 | 2.58 | 15.73 |

As previously described [9], the change as a result of thermal modification may be explained by a darkening of lignin components alone, whereas results here indicate some additional processes, albeit seemingly differing between spruce and beech for bicine and tricine, respectively. This suggests that some form of additional chemical reaction assumed to be a Maillard process has occurred as a result of the combination of treatments, with a $p$-value for the interaction effect of heat treatment and choice of respective chemicals of 0.021, which was possibly as a result of the formation of polymerisation products such as melanoidins [24].

Based on the results in Tables 2 and 3, it would appear that the combined effect of chemical treatment and thermal modification had differing impacts on the beech and spruce, respectively. For softwoods, the treatment with bicine produced the most significant colour change on thermal modification ($p = 0.068$ for S_Bi, but $p < 0.001$ for S_Bi_HT), whilst tricine was most significant with thermally treated beech. The result for bicine and heat treatment for spruce would appear to confirm the retention of material suggested from weight uptakes in Figure 2, given that the retained materials show greater colouration, typical of a Maillard-type reaction occurring.

### 3.2.2. Contact Angle Measurements

Studies into how treated beech and spruce samples adsorb a droplet of water over a period of time are shown in Table 4. It can be seen that the thermal modification, albeit under mild conditions, had a significant effect on how quickly a droplet was adsorbed into the wood surface. Increasing the thermal treatment temperatures from those in this study (165 °C) to typical thermal modification conditions (above 190 °C) would further increase the hydrophobicity of the wood surface and help maintain the droplet stability. However, operating the thermal processing at these higher temperatures would result in thermal decomposition of the bicine and tricine.

**Table 4.** Overview of contact angle studies of a water droplet over time. B = beech, S = spruce, Bi = bicine, T = tricine, HT = thermal modification, C = control.

| | | Time (Seconds) | | | | | |
|---|---|---|---|---|---|---|---|
| | | **1 s** | **10 s** | **20 s** | **30 s** | **45 s** | **60 s** |
| Beech | B_C | 70.71 | 57.45 | 53.93 | 51.47 | 48.19 | 45.97 |
| | B_HT | 99.78 | 90.17 | 84.49 | 81.66 | 78.07 | 75.28 |
| | B_Bi | 87.04 | 70.98 | 65.27 | 61.53 | 57.80 | 55.80 |
| | B_Bi_HT | 101.83 | 86.24 | 79.78 | 75.70 | 71.73 | 68.95 |
| | B_Tri | 99.13 | 83.06 | 77.64 | 75.31 | 72.26 | 70.43 |
| | B_Tri_HT | 113.17 | 102.75 | 96.96 | 93.04 | 89.23 | 86.50 |
| Spruce | S_C | 106.05 | 99.87 | 97.99 | 96.31 | 94.74 | 93.86 |
| | S_HT | 112.79 | 110.24 | 109.55 | 109.08 | 108.33 | 107.69 |
| | S_Bi | 86.16 | 69.90 | 63.04 | 59.79 | 58.04 | 56.64 |
| | S_Bi_HT | 102.58 | 87.81 | 82.87 | 79.85 | 76.88 | 75.44 |
| | S_Tri | 102.53 | 86.50 | 81.27 | 77.93 | 74.42 | 72.39 |
| | S_Tri_HT | 108.10 | 101.79 | 99.16 | 97.88 | 96.47 | 95.16 |

It is interesting to note the effect of impregnating bicine and tricine into the wood specimens. For spruce, there is a significant reduction in immediate contact angle, with the droplet being more rapidly adsorbed into the wood surface over the 60-s duration. However, there is a slight increase in droplet stability when beech is treated with bicine ($p < 0.001$). When combined with thermal treatment, bicine treatments are equivalent to the thermal treatment alone, whilst tricine treatment followed by thermal modification resulted in a significant increase in droplet stability (and therefore hydrophobicity) for beech ($p < 0.05$). The combined salt and thermal treatments on spruce yielded results similar to those of the thermal treatment alone. The slightly better results for tricine may be partly described by its different solubility level to bicine (18 g per litre compared to 160 g per litre for bicine). Whilst this did not appear to cause any problems in preparation of the treatment sample, it may be possible that once deposited within the cell wall, the tricine resulted in a coating that slightly enhanced the hydrophobic nature of the wood surface. The potential of subsequent reaction with reduced sugars during the thermal modification process may further enhance this effect.

3.2.3. Dynamic Vapour Sorption (DVS)

The equilibrium moisture contents of samples (Table 5) mainly followed the patterns observed for low temperature thermal modification processes, with the slight decreases noted matching those previously reported under similar conditions [44].

**Table 5.** Average equilibrium moisture content (EMC) values of combined milled samples at 95% relative humidity (RH). B = beech, S = spruce, Bi = bicine, Tri = tricine, HT = thermal modification, C = control.

| **Beech** | **EMC$_{95\% RH}$ (%)** | **Spruce** | **EMC$_{95\% RH}$ (%)** |
|---|---|---|---|
| B_C | 23.40 | S_C | 23.79 |
| B_HT | 19.98 | S_HT | 19.62 |
| B_Bi | 26.95 | S_Bi | 22.96 |
| B_Bi_HT | 25.36 | S_Bi_HT | 20.92 |
| B_Tri | 24.54 | S_Tri | 20.41 |
| B_Tri_HT | 21.58 | S_Tri_HT | 20.86 |

Significant differences were noted in the EMC values for samples treated with bicine and tricine, respectively, for both beech and spruce. For spruce, all samples exhibited a lower EMC than the control sample (with the exception of the samples undergoing heat treatment alone), with these values further lowered following respective heat treatments.

The individual sorption curves show typical patterns for sorption and desorption, though there are slight differences in the hysteresis values of samples, representing the EMC difference at a particular RH. However, as pointed out by Frederiksson and Thybring [45], care must be taken in analysing sorption hysteresis, since the common method, as in this study, depends on the determination of an absorption isotherm followed by desorption. The desorption step, based on the definition by Frederiksson and Thybring [45], is a scanning isotherm, since the isotherm has not been obtained from either a dry or water-saturated state. The hysteresis graphs, shown in Figure 3 for spruce, would seem to suggest that the treatments with bicine and tricine alone resulted in similar results. However, when thermal modification was subsequently undertaken, the hysteresis curve for S_Bi_HT closer matched that of S_HT, whereas the effect of tricine seemed to have the greater effect on the hysteresis results. Similar experiments with beech resulted in almost identical hysteresis curves for all samples related to bicine treatment, whilst for tricine alone there was a similar reduced hysteresis, but when combined with thermal treatment resulted in a curve matching those of the control and thermal treatment, respectively.

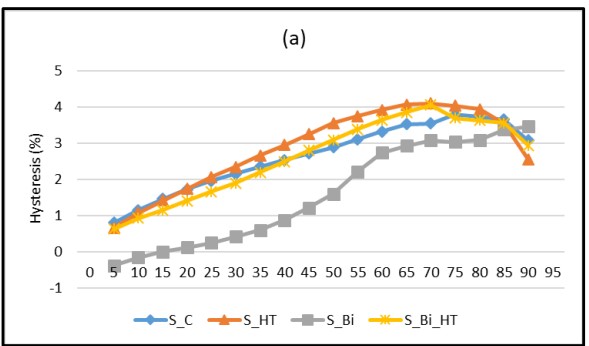 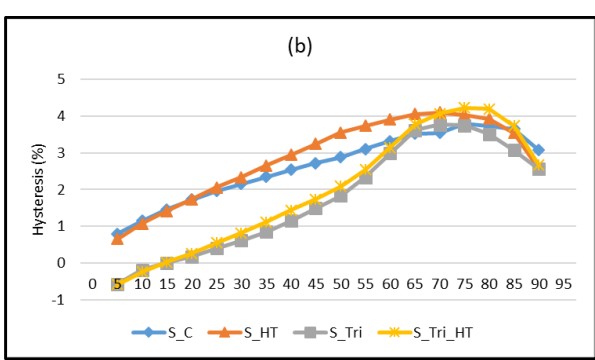

**Figure 3.** Hysteresis results for spruce, showing (**a**) effects of bicine treatment and (**b**) effects of tricine treatment, respectively.

From these results, it would appear there is a reduction in the hysteresis of spruce samples treated with bicine or tricine. This effect is then lost when applying a thermal treatment to the bicine specimens. However, to determine the actual effects, it would be necessary to undertake analysis from water-saturated specimens, as previously suggested [45], instead of running a standard adsorption–desorption analysis between 0 and 95% RHs.

*3.3. Mechanical Testing*

3.3.1. Mechanical Performance Testing

The results of the mechanical tests are shown below (Table 6), in terms of Modulus of Elasticity by three-point bending and compression testing, Modulus of Rupture (MOR) values and Compressive strength (Fm) for both beech and spruce.

For beech, the MOE from three-point bending tests suggested that very little variation was observed, with maximum reductions compared to control samples being around 7.5%. However, compressive MOE values revealed a more significant ($p < 0.001$) reduction, with up to 60% reduction when either bicine or tricine was added. In comparison, the compressive MOE for thermally modified beech only reduced by approximately 5%. MOR values were found to be fairly consistent for samples where thermal modification was not applied. Thus, control samples (B_C) had MOR value of 182.7 N/mm$^2$, whereas treatments with bicine and tricine led to values of 178.4 and 175.4 N/mm$^2$, respectively. Greater changes were noted when looking at thermally modified beech samples, whereby MOR values for samples treated with tricine and subsequently thermally modified (B_Tri_HT) were around 13% lower than for thermal modification alone (B_HT) and samples treated with bicine and thermally modified (B_Bi_HT). In all cases with beech, values of Fm were higher than those of the reference samples, suggesting that the use of bicine or tricine had a positive effect on Fm. Brinell hardness (HB) results failed to show any general trends.



**Table 6.** Overview of mechanical properties of bicine (Bi)- and tricine (Tri)-treated wood. B = beech, S = spruce, HT = thermal modification, C = control. Standard deviation in brackets ().

| | | Three-Point Bending Test | | Compression Test | | |
| --- | --- | --- | --- | --- | --- | --- |
| | **Group** | **MOE (N/mm$^2$)** | **MOR (N/mm$^2$)** | **MOE (N/mm$^2$)** | **Fm (N/mm$^2$)** | **HB (N/mm$^2$)** |
| Beech | B_C | 16,030 (1213) | 182.70 (18.68) | 20,100 (1120) | 99.73 (5.40) | 30.06 (3.45) |
| | B_HT | 14,750 (1732) | 162.60 (34.63) | 19,100 (6140) | 105.43 (7.77) | 29.73 (4.83) |
| | B_Bi | 15,800 (812) | 178.44 (24.74) | 7800 (920) | 108.37 (4.68) | 30.18 (4.66) |
| | B_Bi_HT | 15,956 (1138) | 159.78 (21.90) | 7200 (920) | 106.44 (6.50) | 35.29 (5.52) |
| | B_Tri | 16,120 (2420) | 175.40 (40.32) | 7300 (1060) | 108.24 (5.01) | 35.45 (4.69) |
| | B_Tri_HT | 14,790 (1931) | 140.21 (34.31) | 8100 (740) | 110.84 (9.41) | 30.84 (4.83) |
| Spruce | S_C | 14,410 (1799) | 106.87 (9.72) | 19,400 (970) | 88.73 (11.61) | 16.77 (1.92) |
| | S_HT | 12,019 (4761) | 93.66 (25.51) | 21,000 (1250) | 88.49 (12.38) | 16.74 (2.37) |
| | S_Bi | 12,040 (935) | 97.72 (15.10) | 18,600 (1710) | 96.39 (10.35) | 17.14 (2.97) |
| | S_Bi_HT | 13,440 (1751) | 93.95 (9.62) | 19,000 (1490) | 85.70 (10.10) | 15.77 (2.30) |
| | S_Tri | 12,823 (1860) | 92.93 (32.09) | 8000 (1250) | 88.73 (7.31) | 20.18 (2.39) |
| | S_Tri_HT | 13,805 (1756) | 87.39 (26.73) | 8400 (520) | 97.98 (7.77) | 19.35 (2.18) |

For spruce, MOE values from three-point bending tests showed that thermal modification (S_HT) realised a 14% reduction compared with control samples (S_C). However, direct comparison of the effects of bicine and tricine showed increases in MOE compared with the impregnated samples alone. MOR values generally showed the thermal modification process resulted in lower values to the non-thermally modified analogues. Tukey's HSD Test for compressive MOE values showed that, for spruce, only tricine had a significant effect on values measured ($p < 0.05$). Fm values showed no clear trends, though the results for tricine and thermal modified samples (S_Tri_HT) were considerably higher than the non-thermally modified samples. Similarly, the effects of tricine (both S_Tri and S_Tri_HT) realised higher HB values than any of the other treatments considered herein.

3.3.2. HEMI Tests

Earlier work [46,47] suggested that the main benefit of HEMI testing was for identifying the effects of treatments on the brittleness of materials. Other than ANOVA showing the obvious statistical significant difference between samples undergoing heat treatment and those that did not, results obtained for beech (Table 7) and spruce (Table 8) showed that there were no clear benefits of any specific treatment when considering the resistance to impact milling (RIM). However, there appeared to be a slight influence resulting from treating spruce with tricine (both as a stand-alone treatment and in conjunction with thermal treatment) and with treatment of beech with tricine and heat treatment (S_Bi was shown to be statistically different from ANOVA analysis when considering resultant F values, with $p < 0.001$). The effect of reduced temperature heat treatment showed little effect on the samples, as would be expected, since studies [46] showed that increasing the thermal modification temperature had no significant effect on the RIM values. These similarities in RIM values also suggested no brittleness as a result of treating with bicine or tricine, respectively.

**Table 7.** HEMI test results for beech (B). Bi = bicine, Tri = tricine, HT = thermal modification, C = control.

| | **B_C** | **B_HT** | **B_Bi** | **B_Bi_HT** | **B_Tri** | **B_Tri_HT** |
| --- | --- | --- | --- | --- | --- | --- |
| F (s.d.) | 0.89 (0.23) | 1.55 (0.18) | 0.82 (0.22) | 1.82 (0.69) | 1.41 (0.64) | 1.57 (0.62) |
| I (s.d.) | 55.63 (2.39) | 50.44 (1.80) | 57.24 (1.93) | 53.87 (4.10) | 55.37 (3.85) | 55.22 (2.64) |
| RIM (s.d.) | 88.24 (0.72) | 86.45 (0.43) | 88.70 (0.41) | 87.10 (1.41) | 87.79 (1.35) | 87.63 (0.90) |

**Table 8.** HEMI test results for spruce (S). Bi = bicine, Tri = tricine, HT = thermal modification, C = control.

|  | S_C | S_HT | S_Bi | S_Bi_HT | S_Tri | S_Tri_HT |
|---|---|---|---|---|---|---|
| F (s.d.) | 2.49 (0.66) | 3.65 (0.60) | 3.08 (0.68) | 3.79 (0.98) | 4.04 (1.13) | 5.63 (1.26) |
| I (s.d.) | 28.65 (4.55) | 27.86 (1.73) | 29.15 (4.68) | 24.32 (2.61) | 29.64 (4.64) | 25.86 (3.94) |
| RIM (s.d.) | 80.30 (1.33) | 79.22 (0.66) | 79.98 (1.48) | 78.24 (1.17) | 79.38 (1.86) | 77.24 (1.74) |

In all the treatments, it can be seen that the percentage of fine material present after the HEMI test was higher following the application of heat treatment ($p < 0.05$), any minor change in I values could result in greater fragmentation to smaller pieces.

### 3.4. Chemical Tests

3.4.1. VOC Analysis

Qualitative analysis of samples of collected extractives by gas chromatography–mass spectrometry (GC-MS) showed, as expected, the untreated controls exhibited typical VOCs: 3-carene, α-pinene, β-pinene, α-thujene, β-terpinene and formic acid in both beech and spruce, with an additional peak for acetaldehyde present in spruce. Table 9 gives an overview of the major VOCs detected from samples. Results from experiments combining bicine and heat treatment showed a large number of minor peaks, which could not be assigned probable structures.

**Table 9.** Principal compounds detected by GC-MS of volatile organics from beech (B) and spruce (S) reactions involving bicine (Bi) and tricine (Tri). HT = thermal modification.

| VOC Detected | Formula | Mol. Wt. | B_Bi | B_Bi_HT | B_Tri | B_Tri_HT | S_Bi | S_Bi_HT | S_Tri | S_Tri_HT |
|---|---|---|---|---|---|---|---|---|---|---|
| Formic Acid | $CH_2O_2$ | 46.03 |  |  |  |  | √ |  |  |  |
| Acetaldehyde | $C_2H_4O$ | 44.05 | √ | √ | √ | √ | √ | √ | √ | √ |
| Tetraacetyl-d-xylonic nitrile | $C_{14}H_{17}NO_9$ | 343.29 | √ | √ |  |  | √ | √ |  |  |
| 3N-(7-acetamido-[1,2,4]triazolo [4,3-b][1,2,4]triazol-3-yl)acetamide | $C_7H_9N_7O_2$ |  | √ |  |  |  | √ | √ |  |  |
| Deoxyspergualin | $C_{17}H_{37}N_7O_3$ | 387.50 | √ |  |  |  | √ | √ |  |  |
| Tetrahydro-4h-pyran-4-ol | $C_5H_{10}O_2$ | 102.13 | √ |  |  |  | √ | √ |  |  |
| Oxiranemethanol | $C_3H_6O_2$ | 74.08 | √ | √ | √ | √ | √ | √ | √ | √ |
| O-methylisourea | $C_2H_6N_2O$ | 74.08 | √ | √ | √ | √ | √ | √ | √ | √ |
| Propane | $C_3H_8$ | 44.10 | √ | √ | √ | √ | √ | √ | √ | √ |

From results shown in Table 9, there appears to be more indication of reactions using bicine, though it is uncertain how identified compounds such as tetraacetyl-d-xylonic nitrile, 3N-(7-acetamido-[1,2,4]triazolo[4,3-b][1,2,4]triazol-3-yl)acetamide and deoxyspergualin can be derived, particularly from impregnation with both spruce and beech, where a maximum temperature of 60 °C should have limited any fragmentation of the reagent, whilst being below anticipated temperatures for reactions to occur. Despite this, it is probable that that tetraacetyl-d-xylonic nitrile is a result of an interaction between the chemical reagents and a hemicellulose fragment, whilst the presence of tetrahydro-4h-pyran-4-ol could also result from the breakdown of the hemicellulosic components in wood, though it is interesting that this only occurred at low temperatures in bicine-treated beech, and for the bicine treatment and bicine-high temperature treatment of spruce.

The presence of all the components identified to date did not provide any additional insight into possible reaction mechanisms at this time, other than the likelihood of probable incorporation of nitrogenous moieties from the bicine or tricine into hemicellulosic fragments.

### 3.4.2. Infrared Spectrometry

A detailed investigation into the FT-IR spectrometry of samples has previously been reported [48]. This showed that both the infrared spectra and their derivatives for the control (S_C and B_C) and the thermal treated (S_HT and B_HT) samples did not indicate any significant differences occurring as a result of the thermal process. However, the spectra for treatments involving bicine or tricine showed the presence of the chemicals, though the peaks were found to overlap with those from the wood at higher wavenumbers (3700 to 2500 $cm^{-1}$).

Analysis at lower wavenumbers, along with the respective second derivative spectra, offered more insight into possible interactions. These results allowed the observation of the peaks with bicine treatments described in Table 10, based on spectra shown in Figure 4. Comparative peaks from tricine experiments were less pronounced, probably due to the lower weight uptake of samples.

The use of chemometric methods such as principal component analysis and hierarchal cluster analysis [48] suggested the occurrence of reactions between the functional groups from bicine and tricine and those present in wood structure. In the case of bicine, this could suggest that a chain displacement from bicine to create the Amadori-type products and activation of the N via a quaternary ammonium intermediate is typical of a Maillard reaction. However, assessment of the spectra alongside VOC analysis did not give conclusive evidence of such mechanisms.

**Table 10.** Selection of peaks showing significant differences in 2nd derivate FTIR spectra for reactions involving bicine.

| Wavenumber ($cm^{-1}$) | Peak Identification |
| --- | --- |
| 1740 | Carbonyl/carboxyl stretching vibration |
| 1647 | Carbonyl/carboxyl stretching vibration |
| 1493 | Methyl or methylene deformation vibration |
| 1465 | Methyl or methylene deformation vibration |
| 1422 | Methyl or methylene deformation vibration |
| 1402 | OH stretching |
| 1317 | OH deformation |
| 1267 | C-N stretching vibration in amine groups |
| 1208 | C-N stretching vibration in amine groups |
| 1166 | C-O groups stretching vibration |
| 1118 | C-O groups stretching vibration |
| 1074 | C-O groups stretching vibration |
| 1045 | C-O groups stretching vibration |
| 1024 | C-O groups stretching vibration |

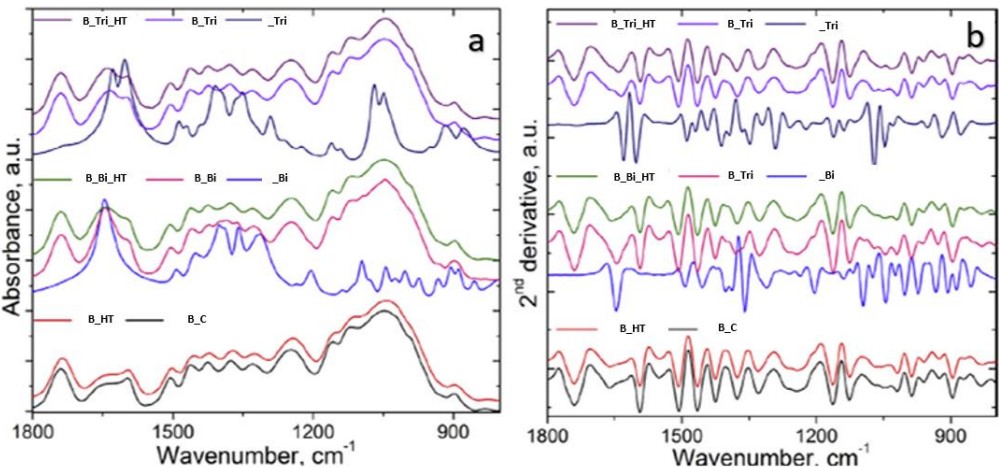

**Figure 4.** *Cont.*

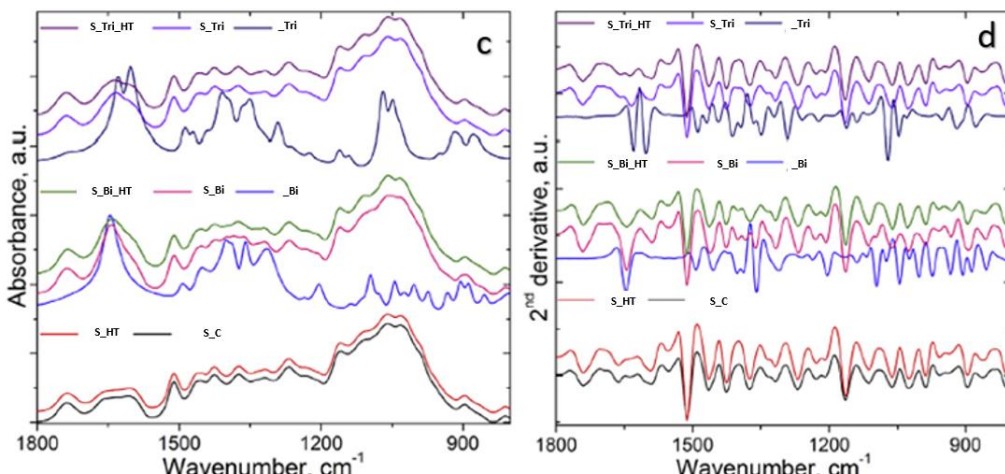

**Figure 4.** FTIR and 2nd derivative spectra of beech and spruce samples. (**a**) FT-IR spectra beech samples; (**b**) 2nd derivative spectra for beech samples; (**c**) FT-IR spectra spruce samples; (**d**) 2nd derivative spectra for spruce samples.

### 3.5. Biological Efficacy

### 3.5.1. Effects against Fungal Decay

Results from experiments with bicine and tricine, both individually and in combination with a reduced thermal modification, did not provide any discernible variation in effects against the three fungal species studied in these trials. However, there were significant variations in degradation studies depending on undertaking the prerequisite EN84 accelerated weathering prior to decay [41]. This is clear from Figure 5 for beech and Figure 6 for spruce, respectively, outlining analysis of decay with *Gloeophyllum trabeum*, *Poria monticola* and *Trametes versicolor*, respectively.

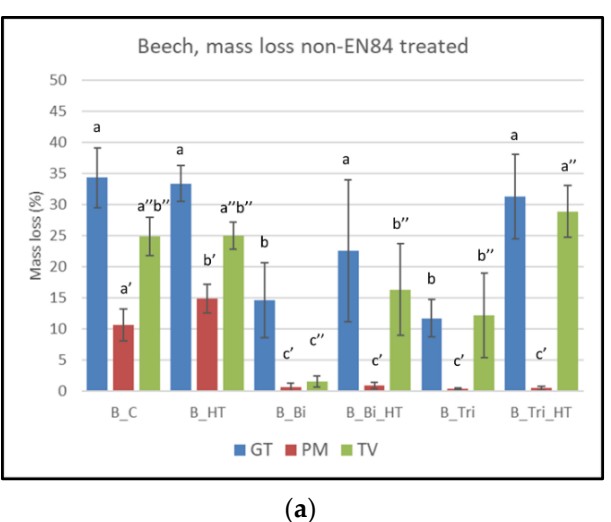

(**a**)

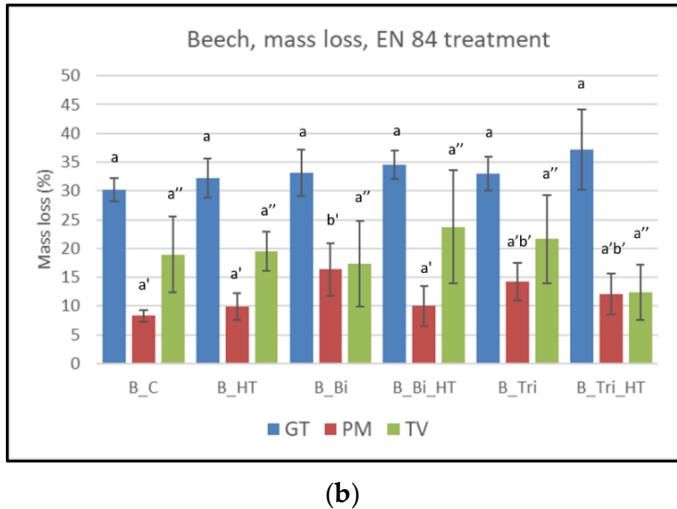

(**b**)

**Figure 5.** Mass loss of beech as a result of fungal decay, (**a**) without EN84 weathering; (**b**) with EN84 weathering. GT—*G. trabeum*, PM—*P. monticola*, TV—*T. versicolor*. Tukey analysis groups shown above each bar graph.

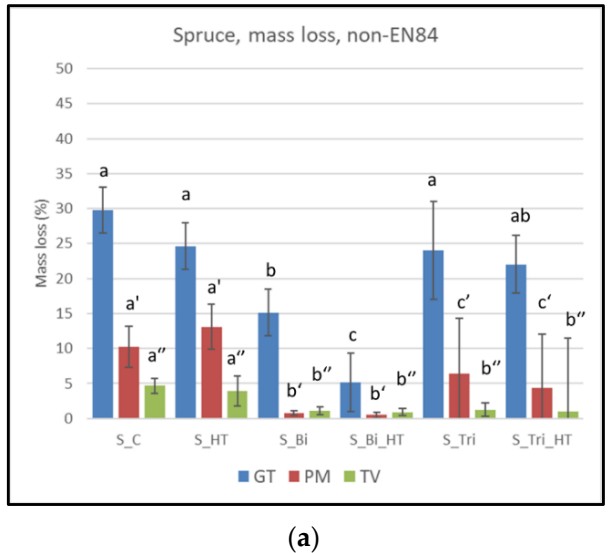

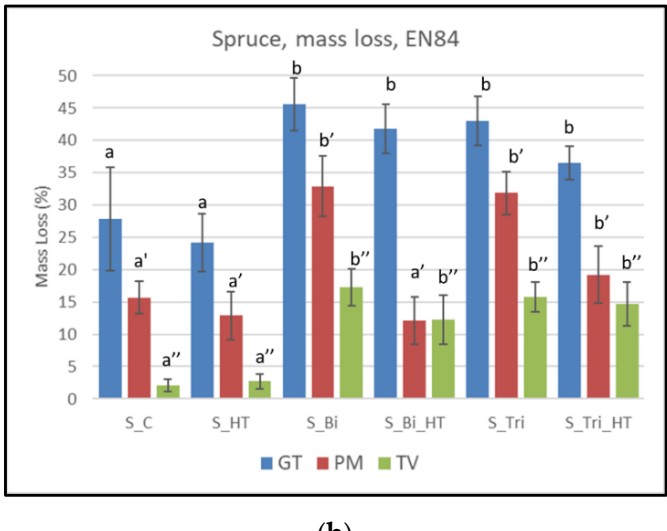

(**a**) (**b**)

**Figure 6.** Mass loss of spruce as a result of fungal decay, (**a**) without EN84 weathering; (**b**) with EN84 weathering. GT—*G. trabeum*, PM—*P. monticola*, TV—*T. versicolor*. Tukey analysis groups shown above each bar graph.

ANOVA analysis of results of non-EN84 accelerated ageing of both beech and spruce showed there were significant differences for all the fungi ($p < 0.05$). From Figures 5 and 6, it can be seen that the EN84 accelerated ageing of samples had a significant effect on the effectiveness of treatments considered in this study. Whereas the spruce control (S_C) and beech controls resulted in similar levels of decay irrespective of EN84, values for other treatments varied considerably, with the greatest effects noted for treatments of both spruce and beech involving bicine or tricine in non-EN84 tests, particularly when assessing with *P. monticola*. However, decay rates were similar or worse where EN84 ageing was carried out prior to decay studies. Similarly, when spruce underwent decay with *G. trabeum* with bicine (S_Bi and S_Bi_HT) without EN84, significant decreases in decay were noted, though these were found to result in greater decay rates when EN84 was carried out. A similar effect was noted with beech (B_Bi).

In an attempt to ascertain why the non-EN84 treated samples experienced considerable reduction in decay levels, moisture content assessments were also carried out on these samples. Results (Figures 7 and 8) suggest that the presence of bicine or tricine as part of the treatment regime significantly affected the moisture content levels noted for the various experimental regimes. Such increases were, however, not noted when looking at equilibrium moisture content measurements (Table 5). The exceptionally high moisture contents noted for some of the non-EN84 treated samples during fungal decay tests may result in water logging of samples, which in turn diminishes the fungal activity during the trial. However, these high moisture contents appear to be within the functional limits for the fungal species used in these trials. This suggests that the presence of the non-leached treatments may have a direct effect on the degree of biological attack by selected fungi.

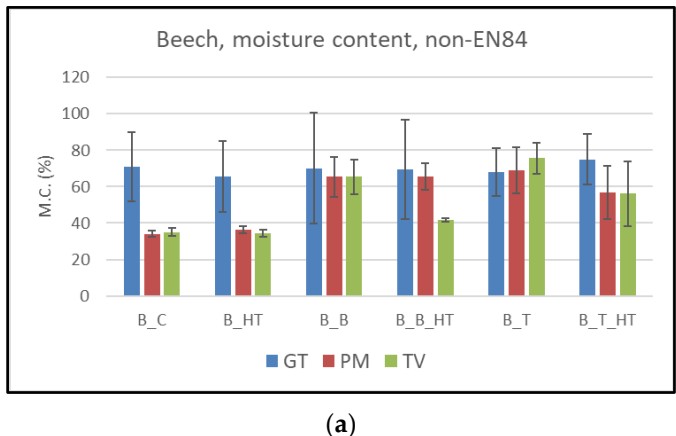
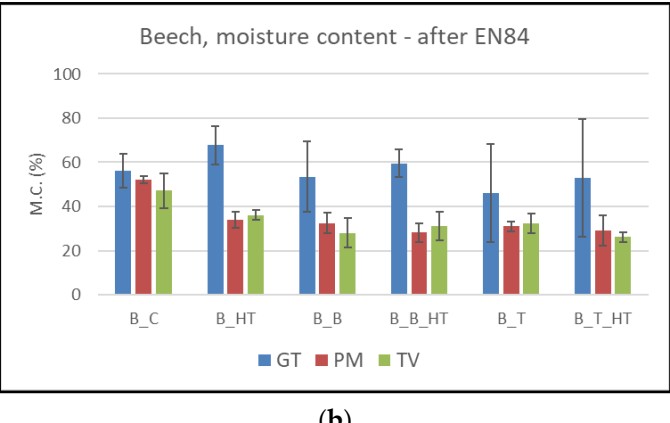

**Figure 7.** Moisture content of beech measured during fungal decay, (**a**) without EN84 weathering; (**b**) with EN84 weathering. GT—*G. trabeum*, PM—*P. monticola*, TV—*T. versicolor*.

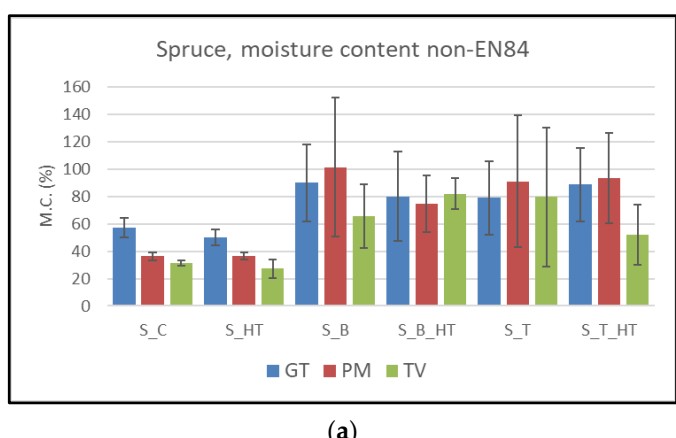
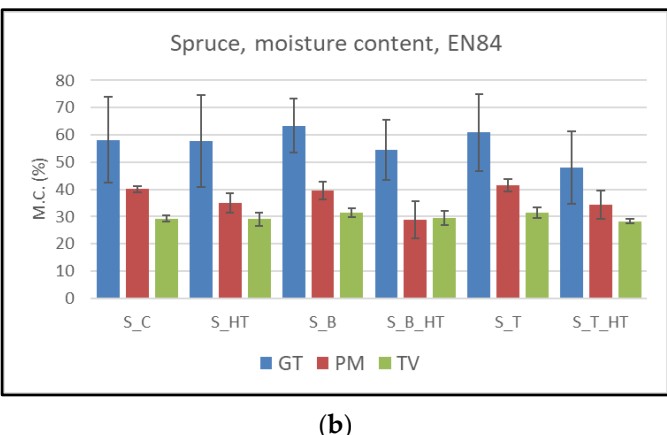

**Figure 8.** Moisture content of spruce measured during fungal decay, (**a**) without EN84 weathering; (**b**) with EN84 weathering. GT—*G. trabeum*, PM—*P. monticola*, TV—*T. versicolor*.

### 3.5.2. Termite Testing

As shown in Figure 9, the survival of the termites and the mass loss showed some differences between the treatments, although a high variability of the results was observed for some treatments. The durability of thermal modified wood is recognised as low and the results of the present work are in accordance with that perception. This low durability fits in well with previously reported data [49,50].

When considering the degree of termite attack of treated samples, it was found that S_B_HT had the lowest attack grade (2.7), followed by S_B, S_Tri and B_Bi, each of which had a grade of 3.0. The treatment B_Tri was found to have a very slight improvement (3.7) over control samples, which, along with the remaining treatments, had a grade of 4.0. In terms of termite survival rate, termites exposed to beech (untreated control and treated wood), though with high variability of results, did not show significant differences among them (F = 1.83, $p$ = 0.181). For spruce wood (untreated control and treated wood), significant differences (F = 6.28, $p$ = 0.004) were observed. Termites exposed to spruce treated with Bi_HT and Tri scored significantly lower survival rates than termites that fed on untreated controls (both pine and spruce) and heat-treated spruce (S_HT), with specific $p$ values shown in Table 11.

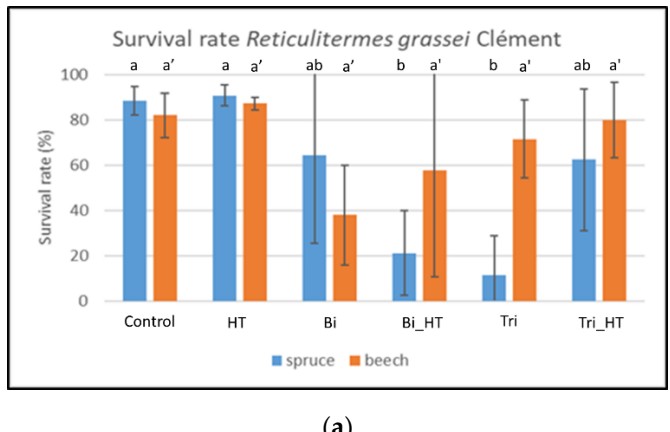

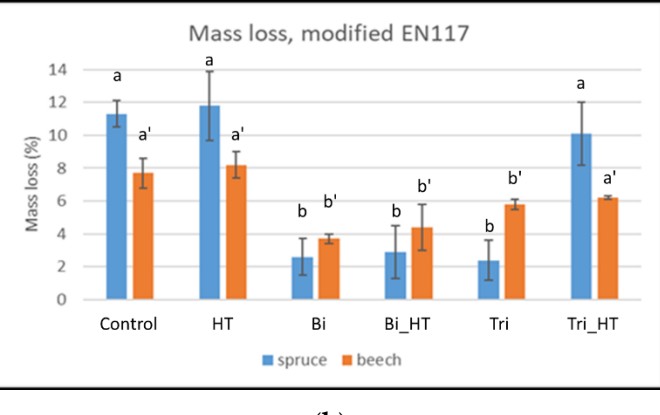

(**a**)                                  (**b**)

**Figure 9.** Overview of termite tests with various treatments. (**a**) Survival rates of *Reticulitermes grassei* Clément and (**b**) mass losses of samples following a modified EN117 experiment [42]. Tukey analysis groups shown above each bar graph.

**Table 11.** Significant results ($p < 0.05$) of the Tukey's honestly significant difference test (HSD) for termite survival in untreated and treated spruce (non-significant $p$ values are not shown in this table).

|  |  | _HT | _Bi | _Bi_HT | _Tri | _Tri_HT | S_C |
|---|---|---|---|---|---|---|---|
| Heat treatment | _HT | x |  |  |  |  |  |
| Bicine and oven dry | _Bi |  | x |  |  |  |  |
| Bicine, oven dry and heat treatment | _Bi_HT | 0.028 |  | x |  |  |  |
| Tricine and oven dry | _Tri | 0.012 |  |  | x |  |  |
| Tricine, oven dry and heat treatment | _Tri_HT |  |  |  |  | x |  |
| Spruce control | S_C |  |  | 0.035 | 0.015 |  | x |

For mass loss, termite consumption was significantly different among beech (untreated control and treated wood) (F = 14.40 $p < 0.001$); termites consumed less B_Bi and B_Bi_HT when compared with control and B_HT (all $p < 0.001$). The same significant differences were only found for tricine-treated beech without HT ($p = 0.031$). For spruce, significant differences among untreated and treated spruce were also observed (F = 26.96, $p < 0.001$); wood treated with S_Bi and S_Tri showed significantly lower mass losses relatively to the group of untreated spruce controls (all $p < 0.001$). S_Bi_HT-treated wood showed significantly lower mass loss than untreated control of spruce, heat treated spruce and THT ($p < 0.001$).

In addition to these mass loss data, a more detailed investigation into the effects of bicine and tricine as a result of the treatments detailed herein on the digestive protists within termite guts has been published [51]. Whilst both bicine and tricine are recognised as Good's buffers and have similar properties (e.g., pH levels), it is interesting to compare similar treatments between the two wood species. When bicine is introduced without subsequent thermal treatment, termite survival is considerably lower for beech than spruce. However, when subsequent thermal treatment was applied, termite survival rates were lower in spruce. This could suggest some mode of action between the bicine and pine as a result of the thermal treatment and not just the effect of the chemical alone.

Given the fact that EN84 treatment was not undertaken prior to exposure of these tests, there may be an increased impact of the unleached bicine or tricine within the wood samples having a direct effect on termite feeding habits and potential toxic effects due to reductions in specific protists within the termite guts [51].

## 4. Conclusions

The use of bicine and tricine as part of an enhanced thermal modification process has been considered. The results herein were part of establishing the feasibility of such

combined chemical/thermal modification processes for treating wood. However, the thermal stability of the selected compounds resulted in the need for a reduced thermal modification temperature, which would have expected impacts on the effectiveness of the thermal modification on its own.

Studies using colour, FT-IR spectroscopy and VOC analysis seem to suggest there is some level of chemical interaction between the treatments and the wood resulting from the trials undertaken, particularly through the presence of compounds such as tetraacetyl-d-xylonic nitrile, though no clear evidence of exactly what the mechanism is has been determined to date. The use of principal component analysis with FT-IR studies confirmed specific peaks were a direct result of chemical reactions.

The hypothesis of combining chemical and thermal reactions of wood resulting in equivalent or better mechanical properties were not borne out in these studies, with three-point bending and compression tests being reduced. There were slight improvements in some Brinell hardness data, though this may be a direct result of the increased density resulting from the uptake of bicine or tricine. The lack of improvements may also be a direct result of the thermal instability of the bicine and tricine.

The results from fungal and termite attack of samples suggested that significant improvements were possible for experiments where the EN84 weathering had not been undertaken. Whilst moisture content levels for fungal decay tests on non-weathered samples suggested the possibility of water logging occurring, corresponding samples tested under standard hysteresis conditions did not show such significant uptakes of moisture. When samples underwent EN84 weathering prior to testing, the beneficial effects were not noted. This indicated that any benefits were lost when the bicine or tricine was leached from the samples. Thus, it would seem logical that further means of entrapment within the wood cell wall structure needs to be considered in future work.

Despite the inconclusive results from this study, there are sufficient indications to suggest some degree of reaction is occurring. Part of the issue with the process undertaken in this study was the use of an open system, whereby any potential intermediate moieties resulting from Maillard-type reactions were volatilised before subsequent reactions could occur. This could potentially be overcome through the use of a closed system reactor, thereby allowing these intermediate groups to undergo further reactions. In addition, the risk of thermal degradation of the key reagents in this study (bicine and tricine) at temperatures typically employed for more effective thermal modification (between 180 and 210 °C) may limit the degree of reactivity encountered in this study, even though reduced heat treatment temperatures (160 °C) were employed. Further consideration into the use of reagents that are more stable at elevated temperatures may enable the full effect of Maillard-type reactions to be explored in more detail.

**Author Contributions:** In this manuscript, the activities of the following persons are hereby summarized: conceptualization, D.J., M.H. (Miha Humar) and D.K.; methodology, D.K., M.H. (Miha Hočevar), C.-M.P., C.B. and L.N.; validation, M.H. (Miha Humar), A.Z., M.-C.P. and G.O.; formal analysis, D.J., D.K. and M.H. (Miha Humar), S.F.C.; investigation, D.J., M.H. (Miha Humar), L.N. and C.-M.P.; writing—original draft preparation, D.J.; writing—all authors; visualization, D.J. and M.H. (Miha Humar); statistical analysis, L.N.; supervision, D.J. and D.S.; project administration, D.J. and D.S.; funding acquisition, D.J. All authors have read and agreed to the published version of the manuscript.

**Funding:** This research was funded by COST Action FP1407 "Understanding wood modification through an integrated scientific and environmental impact approach (ModWoodLife)", which provided a Short-Term Scientific Mission (STSM) grant to D.J. to initiate this work.

**Institutional Review Board Statement:** Not applicable.

**Informed Consent Statement:** Not applicable.

**Data Availability Statement:** Not applicable.

**Acknowledgments:** The involvement of several co-authors as a result of discussions and interaction at both COST FP1407 "Understanding wood modification through an integrated scientific and environmental impact approach (ModWoodLife)" and COST FP1303 "Performance of bio-based building materials" is hereby acknowledged. Additional support to D.J. and D.S. through the project "Advanced research supporting the forestry and wood-processing sector's adaptation to global change and the 4th industrial revolution", OP RDE (Grant No. CZ.02.1.01/0.0/0.0/16_019/0000803) and CT WOOD—a centre of excellence at Luleå University of Technology supported by the Swedish wood industry—is also gratefully acknowledged (D.J. and D.S.). The termite work (L.N.) was conducted under LNEC P2I project ConstBio. Support of the Slovenian Research Agency (D.K., M.H. (Miha Hočevar), M.H. (Miha Humar) and A.Z.) in the frame of the programme P4-0015 is acknowledged as well.

**Conflicts of Interest:** There are no conflict of interest.

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
