# Peer review of "Evaluation of the Effect of a Combined Chemical and Thermal Modification of Wood though the Use of Bicine and Tricine"

_forests, doi:10.3390/f13060834_

Round 1
Reviewer 1 Report
This paper evaluates the effect of thermal and chemical modification in wood. This is an important topic and fits the scope of the journal. The paper presents a series of tests, which were used to see if heat treatment (HT) combined with bicine and tricine would improve the properties of wood. However, in most of the tests HT did better alone than when combined with chemical treatments, which shows this study needs further investigation/experiments. There too many tests and it is hard to follow the discussion, and maybe they should separate physical/mechanical and biodegradation properties, or at least present in a way that is less complicated for the readers. The authors should re-write the article and present the results as an ongoing research and as preliminary study, since this paper does not present any conclusive results. In the conclusion they should present how they will solve current pitfaults, what are the next steps for this research and the additional experiments to improve the methodology proposed in this study.
Author Response
Many thanks for the constructive comments for the manuscript. The experiments have been grouped into sections as suggested, so we hope this makes this a bit easier to follow. Additional information has been added into the conclusions explaining why the reactions were not as obvious as expected and how these may be overcome in further work. IN response to another reviewer, we have also added anova statistical analysis where possible as a means of better showing significance of individual reactions. On behalf of the authors, I hope these changes are to your satisfaction and allow this paper to progress through to publication stage.
Kind regards
Dennis Jones
Reviewer 2 Report
Review of the article: “Evaluation of the effect of a combined chemical and thermal modification of wood though the use of bicine and tricine”
The topic of this research and output might be useful for the Forests readers. The conducted work is worth to publish, however in order to improve the manuscript the following suggestions should be considered.
Abstract
In my opinion, the abstract contains a lot of valuable information, but is too long. In this section, the most important dependencies should be presented, and the rest should be listed in the section Conclusions.
Line 28: Please enter a full name i.e. Picea abies (L.) H.Karst, Fagus sylvatica L. instead of Picea abies and beech Fagus sylvatica, respectively.
Introduction
Line 78: …relative humidity… – it means relative air humidity? during wood heat treatment? Please clarify.
Lines 78-81: The sentence is too long. Please divide into two.
Lines 88-89: …to improve the dimensional stability, decay resistance and water performance of wood, and better mechanical properties [21].
- to improve better mechanical properties? This phrase is incomprehensible in the sentence.
Materials and Methods
Line 160: Note as for line 28.
Specimen treatment
Please include the following information in your article:
- The wood material should be well characterized, i.e.
- where it came from,
- which part of the trunk was obtained from wood,
- from which zone (juvenile, mature),
- what was its density.
- You use the same abbreviation for beech (B) and bicine (B). It seems to me that in order to avoid any understatements and misinterpretation, beech wood should be called BW for short. Or it should be written that the first symbol is for wood and the second is for bicine.
- What were the dimensions of the samples?
- How many samples were used to modify a given species?
- In Table 1 you use the abbreviation HT for heat treatment, in the text TM for thermal modification (Line 176). Please standardize the record.
Colour analysis
- Line 186: …CIE Lab system… You should write …CIE L*a*b* system… Please standardize the record. In 194 Line you wrote CIE L*a*b* system.
Contact angle measurements
- What reference liquid was used? Water, diiodomethane, formamide? The characteristics of the liquid must be given.
- Gindl, M.; Sinn, G.; Gindl, W.; Reiterer, A.; Tschegg, S., 2001: A comparison of different methods to calculate the surface free energy of wood using contact angle measurements. Colloids and Surfaces A – Physicochemicaland Engineering Aspects, 181 (1-3): 279-287- Laskowska A., Kozakiewicz P. 2017: Surface wettability of wood species from tropical and temperate zones by polar and dispersive liquids. Drvna Industrija 68 (4): 299 – 306
Results and Discussion
Figure 3: Figures titles are redundant. Especially that the text of the article mentions beech, spruce and not European beech, Norway spruce. You should add (a), (b) and write: Weight changes as a result of treatments on (a) beech and (b) spruce.
Line 358: You should write ΔL* instead of ΔL.
Table 2: Please standardize the colour parameters according to the comments. Mean values are provided, why are no standard deviations provided?
- Why was no statistical analysis of the research results performed?
Table 3: Mean values are provided, why are no standard deviations provided?- Why was no statistical analysis of the research results performed?
Table 4: Comments as for Table 2. 3.
Table 5: You should add zero e.g. in the data 812 and another like that.
Figure 9: What is the result of such large values of standard deviations i.e. for spruce control?
The study lacks a statistical study of the research results. It is the basis for formulating the right observations and conclusions. Please supplement individual sections of the article with statistical analyzes.

Author Response
Thank for your valuable comments to our manuscript. All factors that were raised have been dealt with accordingly. For section on CIE L*a*b*, a new table including deviations has been included. Furthermore, where possible, anova analysis has been applied. On behalf of the authors, I hope that these changes are to your satisfaction and improve the manuscript to the level where publication is now possible.
Kind regards
Dennis Jones
Reviewer 3 Report
This an excellent written and presented paper. The approach made by the authors is really interesting and innovative. Well done to the great team of authors. I really enjoyed reading this extremely interesting paper.
Author Response
Thank you for your kind comments. Further to requirements of the other reviewers, some changes have been made, specifically shortening the abstract, merging some tests under common themes (physical, chemical, mechanical, biological) and adding some anova statistical analysis of results where appropriate.
Kind regards
Dennis Jones
Round 2
Reviewer 2 Report
The authors made the proposed corrections.
The authors corrected bicine abbreviations to Bi, but in the some tables. Please analyze the figures i.e. Figure 3 and also put the appropriate abbreviations on them, i.e. B_Bi instead of B_B (as in the Table 1). The note also applies to the markings in Table 5.
Author Response
Once again, many thanks for the diligent review of the article. Apologies for not seeing Figure 3, this has been duly changed. Please note that Figure 5 has already been altered, which is more easily noted when looking at the simple markup track changes. On behalf of the authors, we really appreciate you taking the time for proofing the manuscript.